# Interferons and tuft cell numbers are bottlenecks for persistent murine norovirus infection

Somya Aggarwal[1], Forrest C. Walker[1], James S. Weagley[1], Broc T. McCune[2], Xiaofen Wu[1], Lawrence A. Schriefer[1], Heyde Makimaa[1], Dylan Lawrence[1], Pratyush Sridhar[1], Megan T. Baldridge[1,3]*

1 Division of Infectious Diseases, Department of Medicine, Edison Family Center for Genome Sciences & Systems Biology, Washington University School of Medicine, St. Louis, Missouri, United States of America, 2 Department of Microbiology, University of Texas Southwestern Medical Center, Dallas, Texas, United States of America, 3 Department of Molecular Microbiology, Washington University School of Medicine, St. Louis, Missouri, United States of America

* mbaldridge@wustl.edu

**Data Availability Statement:** All data is shown in the main text or supporting information. Barcode

## Abstract

Noroviruses (NoVs) are a leading cause of viral gastroenteritis. Despite global clinical relevance, our understanding of how host factors, such as antiviral cytokines interferons (IFNs), modulate NoV population dynamics is limited. Murine NoV (MNoV) is a tractable *in vivo* model for the study of host regulation of NoV. A persistent strain of MNoV, CR6, establishes a reservoir in intestinal tuft cells for chronic viral shedding in stool. However, the influence of host innate immunity and permissive cell numbers on viral population dynamics is an open question. We generated a pool of 20 different barcoded viruses (CR6$^{BC}$) by inserting 6-nucleotide barcodes at the 3' position of the NS4 gene and used this pool as our viral inoculum for *in vivo* infections of different mouse lines. We found that over the course of persistent CR6 infection, shed virus was predominantly colon-derived, and viral barcode richness decreased over time irrespective of host immune status, suggesting that persistent infection involves a series of reinfection events. In mice lacking the IFN-λ receptor, intestinal barcode richness was enhanced, correlating with increased viral intestinal replication. IL-4 treatment, which increases tuft cell numbers, also increased barcode richness, indicating the abundance of permissive tuft cells to be a bottleneck during CR6 infection. In mice lacking type I IFN signaling (*Ifnar1*$^{-/-}$) or all IFN signaling (*Stat1*$^{-/-}$), barcode diversity at extraintestinal sites was dramatically increased, implicating different IFNs as critical bottlenecks at specific tissue sites. Of interest, extraintestinal barcodes were overlapping but distinct from intestinal barcodes, indicating that disseminated virus represents a distinct viral population than that replicating in the intestine. Barcoded viruses are a valuable tool to explore the influence of host factors on viral diversity in the context of establishment and maintenance of infection as well as dissemination and have provided important insights into how NoV infection proceeds in immunocompetent and immunocompromised hosts.

sequencing data is available at NCBI BioProject (accession number PRJNA1061503).

**Funding:** This work was supported by the National Institutes of Health (NIH) grants R01AI139314 and R01AI127552 (M.T.B.), and the Pew Biomedical Scholars Program of the Pew Charitable Trusts (M.T.B.). F.C.W. was supported by NIH T32GM007067. B.T.M. was supported by NIH F32AI138392. H.M. was supported by NIH R25 GM103757. J.S.W. was supported by NIH T32AI007172. The funders did not play any role in the study design, data collection and analysis, decision to publish, or preparation of the manuscript.

**Competing interests:** The authors have declared that no competing interests exist.

## Author summary

Defining the host factors responsible for controlling viral population dynamics during infection is critical for establishing a thorough understanding of viral transmission, dissemination, pathogenesis, and immune evasion. Here, we employed a barcoded virus strategy to interrogate how host factors modulate viral diversity of CR6, a persistent strain of murine norovirus. By evaluating barcode levels in tissues and stool of wild-type mice, mice lacking critical innate immune response genes, and mice treated with cytokine to enhance susceptible tuft cell levels, we found that both the availability of tuft cells and viral replication limitations imposed by interferon signaling serve as critical bottlenecks for CR6 diversity. Our studies also indicated that stool virus is likely predominantly derived from the colon, and that extraintestinal dissemination of CR6 in immunodeficient mouse strains likely occurs independently of intestinal infection. Our study thus revealed key constraints regulating norovirus population dynamics and provided additional insights into the mechanisms of viral shedding and dissemination.

## Introduction

Transmission bottlenecks can occur when pathogens are transmitted from one infected host to another or even during the spread of infection within an infected host. These bottlenecks are major stochastic forces that can dramatically affect the virulence of rapidly-evolving pathogens like RNA viruses [1,2]. Over the course of infection, viruses encounter both physical barriers and immune pressures within the host that can affect both viral population diversity and the degree of genomic sequence divergence in spatial and temporal manners [3]. Improved understanding of the consequences of individual bottlenecks to viral populations, as well as their combined effects, is critical to both understanding the dynamics of viral pathogenesis and for potentially predicting and limiting emergence of variants of concern.

Human noroviruses (HNoVs) are the leading cause of outbreaks of viral acute gastroenteritis worldwide with an estimated 700 million infections and 200,000 deaths annually [4]. No vaccines or therapies are currently available [5]. HNoVs are single-stranded positive-sense RNA viruses present in three of ten genogroups of the *Caliciviridae* family. In immunocompetent hosts, HNoV causes an acute and generally self-limiting infection, but illness may be more severe and/or chronic in immunocompromised individuals [6,7]. Because viral evolution has been observed in these chronically-infected patients, it has been suggested they may act as viral reservoirs for subsequent emergence of novel HNoVs [8–10]. Defining the selective pressures that regulate establishment and maintenance of NoV populations during chronic infection is thus an important area of inquiry.

Because HNoVs do not robustly infect mice, the discovery of murine norovirus (MNoV) has provided a tractable *in vivo* model for NoV studies [11]. MNoV recapitulates numerous characteristics of HNoV including intestinal replication, fecal-oral transmission, prolonged shedding after infection, genomic organization, and capsid structure [12]. Persistent MNoV strain CR6 serves as a powerful model for asymptomatic chronic viral shedding, and many studies have revealed host factors critical for viral regulation. CR6 exclusively infects tuft cells, chemosensory cells of the intestinal epithelium [13,14], due to their expression of viral receptor CD300LF [15–17]. Tuft cells act as an immune-privileged niche to permit viral persistence [18], and induction of tuft cell hyperplasia by cytokines such as IL-4 and IL-25 consequently promotes CR6 infection [13]. Interferons (IFNs) are also critical for controlling CR6 [19,20]. Type I IFNs (such as IFN-α/β) limit systemic dissemination from the intestine [19,21,22]. In

contrast, type III IFNs (IFN-λ) control intestinal replication; studies in constitutively and conditionally *Ifnlr1*-deficient mice have shown that endogenous IFN-λ acts on tuft cells to limit enteric viral levels and viral shedding [18,21,23,24].

Despite the identification of these regulators of overall viral infection, our understanding of how these selective host pressures affect NoV population dynamics throughout infection is limited. In this study, we infected various strains of mice with genetically marked CR6 carrying short unique nucleic acid sequence "barcodes" [25–29], which permitted quantitative analysis and tracking of viral clones to study viral population dynamics under different conditions.

## Materials and Methods

### Ethics statement

All mice were singly-housed and the experiments were conducted according to the regulations specified by the Washington University Institutional Animal Care and Use Committee under approved protocol 22–0140.

### Construction of barcode library and viral stock generation

To construct the barcoded CR6 library, primers were designed for 6-nucleotide long barcodes to be inserted after nucleotide 2601 of the CR6 genome at the 3' end of NS4 (**S1 Table**). The cleavage site between NS4 (p18) and NS5 (VPg) occurs at $^{870}$E/G$^{871}$ [30], with the barcode insertion of 2 amino acids occurring between amino acids 865 and 866. PCR was performed using Q5 high-fidelity DNA polymerase (NEB; M0491L). The amplicon was treated with KLD enzyme mix (NEB; M0554S) and transformed into competent DH5 alpha *E. coli* cells (Zymo; T3007). Individual colonies were screened for the presence of the barcode by colony PCR and then verified by Sanger sequencing.

Plasmids encoding the barcoded viral genomes were transfected into 293T cells using TransIT-LT1 (Mirus Bio, Madison, WI), and incubated for 48 h at 37˚C. P0 virus was recovered after the freeze-thaw of transfected cells. Clarified 293T supernatants were passaged on BV2 cells at a multiplicity of infection (MOI) of 0.05 in DMEM with 10% FBS to collect P1 virus which was then passaged on BV2 cells in VP-SFM (virus production serum-free media; Gibco; 11681020) containing 1% glutamine and P2 virus was recovered after freeze-thaw of the infected cells followed by centrifugation at $\geq$18,000 × g for 2 min to clear debris. The virus was aliquoted and stored at -80˚C until use. Titers of each barcoded P2 virus were determined by plaque assay and equal plaque-forming units (PFUs) from each barcoded virus were mixed to obtain the pool of barcoded virus, CR6$^{BC}$. For WT CR6, viral stocks were derived from the molecular clone of CR6 as described previously [31].

### Viral growth curves and plaque assays

MNoV growth curves were performed as described previously [17]. Briefly, 5 x 10$^4$ BV2 cells per well were infected in suspension with CR6 and CR6$^{BC}$ at an MOI of 0.05 in 96-well plates. Plates were frozen at 0, 12, 24, and 48 hours post-infection (hpi) and total cell lysate was used in subsequent plaque assays. For plaque assays, BV2 cells were seeded in DMEM with 10% FBS at 2 x 10$^6$ cells/well of a six-well plate and incubated for 16–20 hours at 37˚C. Media was removed, and 10-fold serial dilutions of cell lysate were added to each well for 1 hour at room temperature with gentle rocking. Viral inoculum was removed and 2 mL of overlay media (MEM, 10% FBS, 2mM L-Glutamine, 10 mM HEPES, and 1% methylcellulose) was added. Plates were incubated for 48 hours and then fixed with crystal violet solution (0.2% crystal

violet and 20% ethanol) after removing the overlay media [31]. Plaques were counted and titer was calculated.

## Mouse lines

C57BL/6J wild-type mice were originally purchased from Jackson Laboratories (JAX stock #000664, Bar Harbor, ME) and bred and housed in Washington University animal facilities under specific pathogen, including MNoV, free conditions. Knock-out mice on the C57BL/6J background were maintained in the same conditions and included the following strains: *Stat1*[-/-] (JAX stock #012606), *Ifnar1*[-/-] [32], *Ifngr1*[-/-] (JAX stock #003288, [33]) and *Ifnlr1*[-/-] [34]. *Ifnar1*[-/-]*Ifngr1*[-/-] mice were generated by crossing *Ifnar1*[-/-] and *Ifngr1*[-/-] mice.

## IL-4 treatment

Recombinant IL-4C complexes (rIL-4Cs) were generated as described previously [35]. Briefly, for each mouse 5 μg of murine IL-4 (Peprotech) was mixed with 25 μg anti-IL-4 (Clone 11B11, BioXCell) and incubated for 1–5 min prior to diluting to 200 μl total volume in PBS. rIL-4Cs were administered intraperitoneally in a volume of 200μl twice per mouse at 48 and 24 hours before infection.

## Mouse infection and sample collection

6–9-week-old mice were gavaged with $10^6$ PFUs of CR6$^{BC}$ in a volume of 100μl. Stool (collected directly from each mouse) and tissues were collected at indicated time points post-infection. All stool and tissues were harvested into 2-mL tubes (Sarstedt, Germany) with 1-mm-diameter zirconia/silica beads (Biospec, Bartlesville, OK). Stool and tissues were either processed on the same day or stored at −80˚C.

## RNA extraction and qPCR

RNA extraction from stool and tissues were performed as described previously [34]. Briefly, stool RNA was isolated using QuickRNA Miniprep (Zymoresearch, Irvine, CA) kit and tissue RNA was isolated using TRI Reagent with a Direct-zol-96 RNA kit (Zymo Research, Irvine, CA) according to the manufacturer's instructions. cDNA was synthesized using ImPromII reverse transcriptase system (Promega, Madison, WI) from 5 μL of stool or tissue RNA. Absolute quantification of viral genomes was performed using MNoV TaqMan assays as described previously [36]. PrimeTime qPCR assays were used to quantify expression of *Cd300lf* (Mm. PT.58.13995989) and *Dclk1* (Mm.PT.58.7877738). For normalizing absolute values of viral genome copies or host transcripts from tissues, qPCR for housekeeping gene *Rps29* was used as described previously [37].

## Sequencing and analysis of barcode library

Primers targeting the CR6 genomic regions (Forward primer–TACTGGGACCACGGTTA-CAC; Reverse primer–TCATATTCCTCGTCCGTGAGC) flanking the barcode insertion site were designed with both Illumina adaptor sequences as well as custom indices within forward and reverse primers for demultiplexing. cDNA, synthesized from stool and tissues, was used as the PCR substrate. PCR was performed in reactions containing 18.9 μL RNase/DNase-free water, 2.5 μL 10X High Fidelity PCR Buffer (Invitrogen), 0.5 μL 10 mM dNTPs, 1 μL 50 mM MgSO$_4$, 1.0 μL each of the forward and reverse primers (10 μM final concentration), 0.08 μL Platinum High Fidelity Taq (Invitrogen) and 1.0 μL cDNA. Reactions were held at 94˚C for 2 min to denature the DNA, with amplification proceeding for 32 cycles at 94˚C for 15s, 50˚C

for 30s, and 68°C for 30s followed by a final extension of 2 min at 68°C. Amplicons were pooled and the presence of amplicon confirmed by gel electrophoresis. Small aliquots of PCR products were run on agarose gels and PCR products were pooled in equal amounts as approximated by amplicon band intensity. The pooled products were purified using 0.6X volume AMPure XP magnetic beads to remove primer dimers and unused dNTPs as per manufacturer's instruction. Sequencing was performed on a NextSeq Illumina sequencer (2x150 runs). FASTQ files were demultiplexed and barcodes, plus the surrounding 30 nucleotides of viral genomic sequence to ensure that the barcodes were associated with virus sequence, were extracted from the reads. Barcode counts were enumerated with the "grep" command, with each barcode sequence serving as a search string. Initial data analysis was performed using R [38]. Barcode frequencies are presented as fractions of reads for a specific barcode over total reads containing a viral barcode, with raw read values for each barcode provided in **S2**–**S4 Tables**. The limit of detection for the barcodes was 10 reads, which was derived from within-sequencing-run analysis of naïve mice. Shannon diversity index was calculated using the R package "vegan" [39]. Barcode sequencing data is available at NCBI BioProject (accession number PRJNA1061503).

## Graphing and statistics

Stacked bar plots were created using R package "ggplot2" [40]. The data were analyzed with Prism 10 software (GraphPad Software, San Diego, CA). In all graphs, ns indicates not significant ($p > 0.05$), $^{***} p < 0.001$, $^{**} p < 0.01$, $^{*} p < 0.05$, as determined by Mann-Whitney test, one-way analysis of variance (ANOVA) or Kruskal-Wallis test, or two-way ANOVA with Tukey's multiple-comparison test, as specified in the relevant figure legends.

## Results

### Barcoded CR6 (CR6$^{BC}$) is functionally equivalent to CR6 and maintains barcodes *in vitro* and *in vivo*

Molecularly barcoded virus has been used to study bottlenecks in different virus systems including coxsackievirus, influenza virus, poliovirus, West Nile virus, and Zika virus [26–28,41,42]. To generate barcoded CR6 stocks, 6-nucleotide barcodes were individually inserted at the 3' end of NS4 (at nucleotide 2601) of the CR6 genome (**Fig 1A** and **S1 Table**). Small insertions at this site in the MNoV genome are tolerated, permitting recovery of tagged infectious virus [43]. Twenty distinct tagged viruses were combined at equal plaque-forming unit (PFU) ratios to generate a pooled CR6$^{BC}$ (BC for "barcoded") viral stock. *In vitro* replication of CR6$^{BC}$ in the murine BV2 microglial cell line was equivalent to parental CR6 (**Fig 1B**). Barcode sequencing over 72 hours (h) of *in vitro* growth showed that barcodes were not lost over time (**Fig 1C**), and that the relative proportions of the 20 unique barcodes within the pool remained similar over time (**Fig 1D**). Varying genome copy:PFU ratios were observed amongst the individual barcoded viruses, which may have contributed to initial uneven representation of barcodes within the inoculum (**S1A Fig**). Wild-type (WT) mice were orally inoculated with $10^6$ PFUs of CR6$^{BC}$ or parental CR6 and viral levels were enumerated at 5 days post-infection (dpi) by MNoV-specific qPCR (**Fig 1E and 1F**). These results confirmed no deleterious effect of barcode insertion on either infectivity or replicative capacity *in vitro* or *in vivo*. To confirm that barcode sequences recovered from mice were from replicating virus and not the administered inoculum, we inoculated mice lacking the MNoV receptor CD300LF which are resistant to infection [15–17]. As expected, there was no viral shedding in the stool of *CD300lf$^{-/-}$* mice at

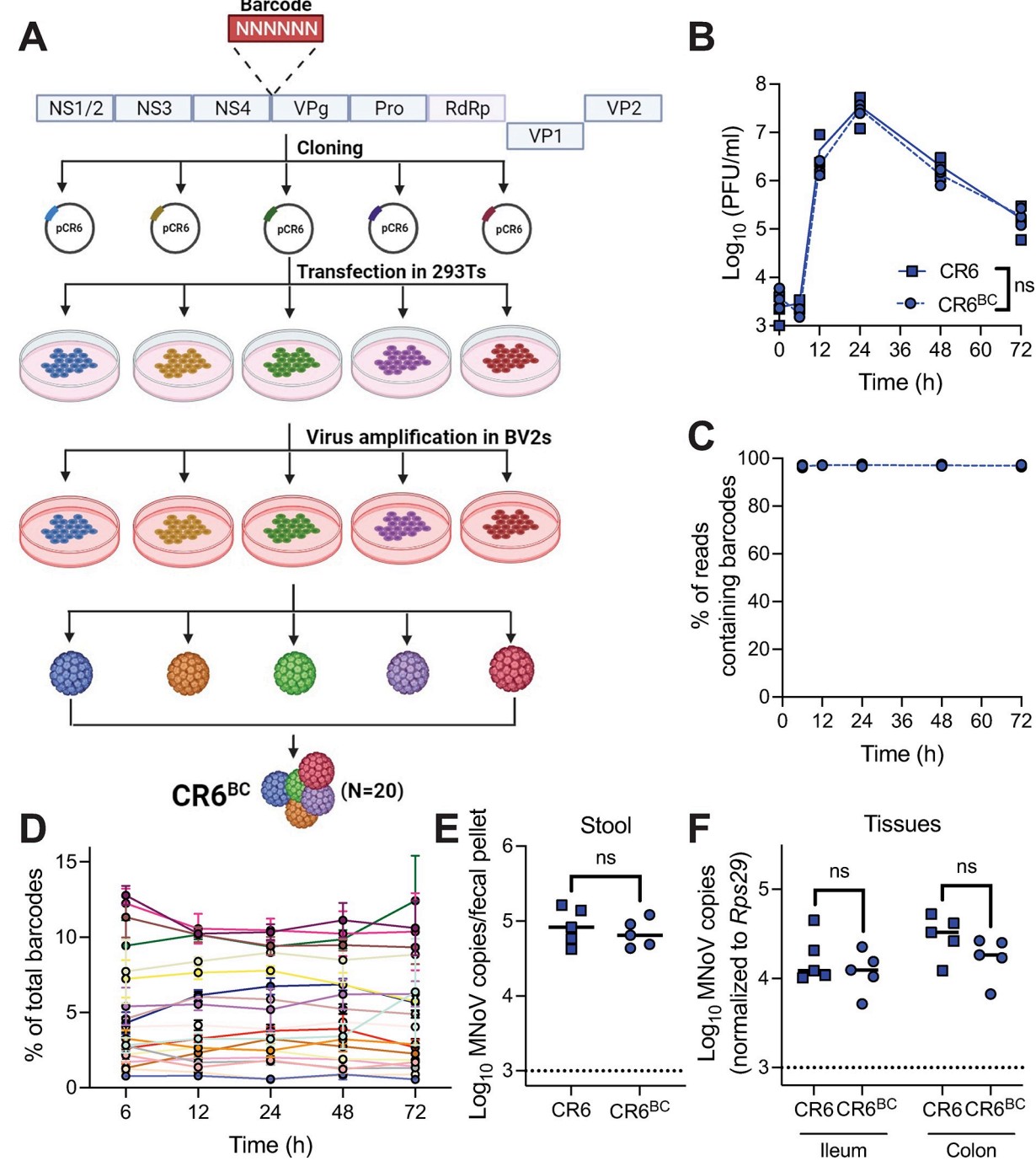

**Fig 1. Barcoded CR6 (CR6^BC) shows similar infectivity to CR6 and maintains barcodes *in vitro* and *in vivo*.** (**A**) Schematic of CR6^BC pool production. Created with BioRender.com. (**B**) Growth curves of parental CR6 and CR6^BC *in vitro* in BV2 cells; data is pooled from three independent experiments. Analyzed by two-way ANOVA. (**C,D**) Proportion of Illumina sequencing reads from growth curve samples from three independent experiments at indicated timepoints containing any barcode (**C**) or each individual barcode (**D**). (**E-F**) Wild-type (WT) mice (N = 5 per group) were orally inoculated with parental CR6 or CR6^BC, and stool MNoV shedding (**E**) and tissue viral levels (**F**) at 5 dpi were assessed by qPCR. Analyzed by Mann-Whitney test. ns = not significant.

3 or 5 days (**S1B Fig**) or virus detected in intestinal tissues at 5 days post-inoculation with CR6$^{BC}$ (**S1C Fig**) and hence no barcode richness was observed (**S1D and S1E Fig**).

## IFN-λ signaling is a bottleneck for intestinal CR6 replication and viral diversity

Endogenous IFN-λ has been well-established to signal through IFNLR1 on tuft cells to limit CR6 replication *in vivo* [18,21,24,34]. To analyze whether IFN-λ also regulates viral population dynamics, we infected WT and *Ifnlr1*$^{-/-}$ mice with CR6$^{BC}$ and collected tissues and stool at 5 dpi (acute timepoint) and 21 dpi (persistent timepoint). Consistent with prior studies [21,23], increased viral shedding was observed in *Ifnlr1*$^{-/-}$ mice at 5 dpi but no difference in shedding was seen at 21 dpi, while increased intestinal tissue viral loads were present in *Ifnlr1*$^{-/-}$ mice at both 5 and 21 dpi (**Fig 2A**).

We next assessed barcode richness in the samples. Of interest, despite the administered inoculum of 20 barcodes, we observed an average of 5.1 barcodes shed in the stool of WT mice even at 5 dpi, with an average of 5.5 and 5.7 barcodes in colon and ileum respectively (**Fig 2B**). Distinct barcodes were observed in each mouse, indicating that this low number was not secondary to poor viability of numerous specific clones (**S2 Table** and **S1F Fig**). This relative paucity of barcodes suggests that a limited number of unique viruses can establish or maintain infection in WT mice at acute timepoints. By 21 dpi, there was a further depletion of barcode numbers with an average of 2.1 stool, 3.5 colon, and 2.5 ileum barcodes (**Fig 2B**), despite similar levels of viral shedding in stool (**Fig 2A**). We found substantially enhanced barcode richness in *Ifnlr1*$^{-/-}$ mice at 5 dpi in both stool and intestinal tissues, but no significant difference between genotypes at 21 dpi (**Fig 2B**). These data indicate that endogenous IFN-λ signaling limits the number of unique viruses able to establish infection in WT mice, serving as an important bottleneck for viral population diversity during acute infection. Enhanced viral shedding and tissue levels in *Ifnlr1*$^{-/-}$ mice (**Fig 2A**) are thus reflective of increased unique productive viral infection events as opposed to just enhanced viral replication following an equivalent number of individual infection events.

In the context of persistent infection, viral population richness is lost in both WT and *Ifnlr1*$^{-/-}$ mice, despite maintenance of barcodes *in vivo* and similar levels of viral shedding in WT mice at 5 and 21 dpi (**Figs 2C and S1G**). It has previously been determined that CR6 exclusively infects short-lived tuft cells [44]. Thus, this loss in population diversity may reflect the requirement for longitudinal reinfection events to maintain persistent infection, or alternately, effects of adaptive immune responses in limiting viral diversity.

## The colon is the dominant source of viral stool shedding

CR6 infects both the small and large intestines and is shed at high levels in the stool. However, whether shed virus is predominantly derived from the small or large intestine, or both, is unclear. To answer this question, we analyzed the specific barcodes in ileum, colon and stool within individual animals to evaluate if they were shared. We found that during acute infection, most barcodes were shared between colon, ileum, and stool in both WT (~33%) and *Ifnlr1*$^{-/-}$ mice (~74%) (**Fig 3A and 3B**), suggesting initial infection events in the intestine contribute to both fecal shedding and infection along the gastrointestinal tract. Whether ileal infection contributes to colonic infection or vice-versa, wherein re-infection of ileal tissue could occur secondary to coprophagy of colon-derived shed virus, is unclear. A subset of barcodes was exclusively detected in colon and ileal tissues but not in stool, or alternately were exclusively found in ileal or colonic tissue alone. These data raise the possibility of infection events at either site that do not contribute to viral shedding in stool, or of

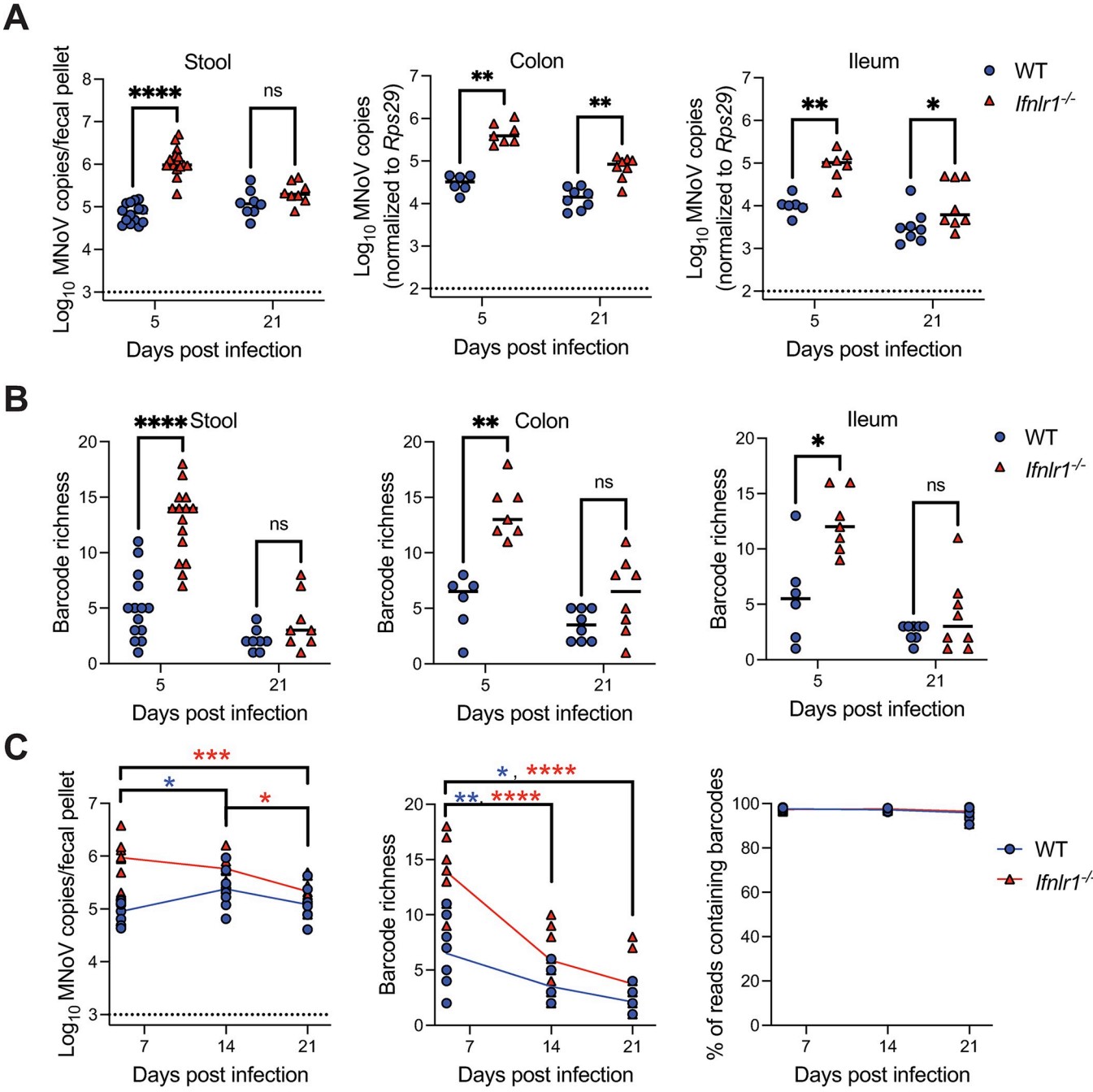

**Fig 2. IFN-λ signaling is a bottleneck for intestinal CR6 replication and viral diversity.** WT or *Ifnlr1*[-/-] mice were inoculated with CR6[BC] and stool and tissues were collected at 5 dpi [WT (stool N = 14; tissue N = 6); *Ifnlr1*[-/-] (stool N = 15; tissue N = 7)], or 21 dpi [WT (N = 8); *Ifnlr1*[-/-] (N = 8)]. (**A**) MNoV genome copies in fecal pellets, colon, or ileum as quantified by qPCR. (**B**) Barcode richness in stool, colon, and ileum. Results were analyzed by Mann-Whitney test from three independent experiments. (**C**) Stool pellets were longitudinally collected at indicated timepoints and MNoV genome copies and barcode richness were assessed, as well as the proportion of reads maintaining barcodes. Results were analyzed by two-way ANOVA with Tukey's multiple comparisons test from three independent experiments. Blue asterisks denote values for WT; red asterisks for *Ifnlr1*[-/-]. *p < 0.05; **p < 0.01; ***p < 0.001; ****p < 0.0001; ns, not significant.

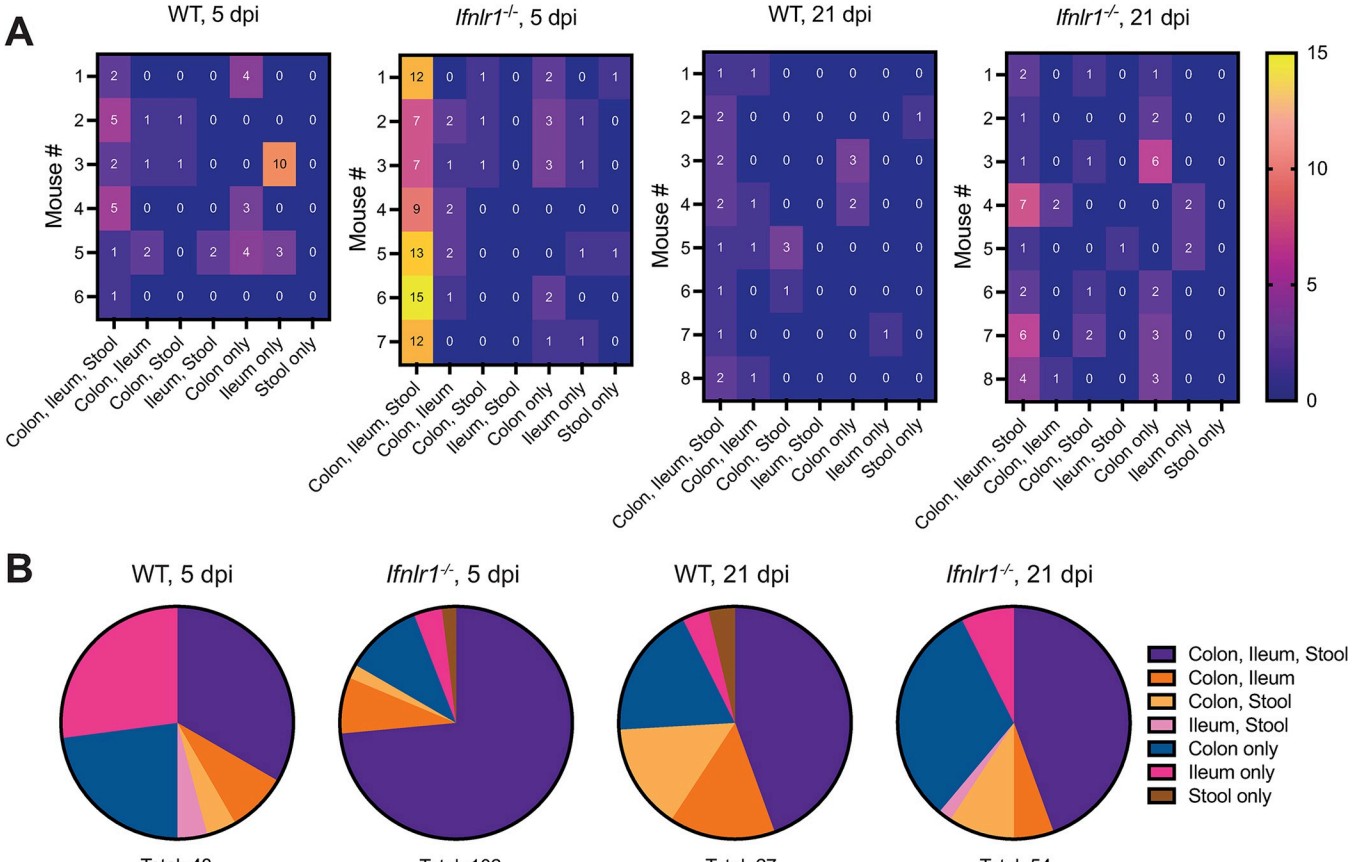

**Fig 3. The colon is the dominant source of viral stool shedding.** Barcode distribution in CR6$^{BC}$-infected WT and *Ifnlr1*$^{-/-}$ mice at 5 and 21 dpi was analyzed for being unique or shared at different sites. (**A**) Heatmaps indicate the number of barcodes detected at the indicated unique or shared sites per mouse. (**B**) Pie-charts indicate the proportions of the total number of barcodes detected at the indicated unique or shared site, summed from all mice in the indicated groups.

variable shedding following these infection events that may not be captured by analysis of single fecal specimens.

At day 21, a timepoint when overall barcode richness was lost (**Fig 2B and 2C**), we observed a similar proportion of barcodes shared between colon, ileum, and stool in WT and *Ifnlr1*$^{-/-}$ mice (~44%) (**Fig 3A and 3B**), with barcodes distributed to a greater degree amongst individual or shared tissues. Across the 5 and 21 dpi timepoints, many barcodes observed in the stool were also present in both colon and ileum, precluding identification of the tissue source of shed barcodes. However, we observed that the likelihood of observing barcodes co-occurring in ileum and stool was significantly lower than observing co-occurrence in colon and stool (Fisher's exact test, p = 0.0070). Thus, while ileum may contribute some shed virus, our data support the colon as the likely dominant source of CR6 viral shedding, and also indicate that throughout infection, the majority of virus is shared between both small and large intestinal tissues.

## Tuft cell abundance is a bottleneck for intestinal CR6 replication and viral diversity

Tuft cells are the exclusive cells infected by CR6 in the intestinal epithelium in the acute setting and also provide an immune-privileged niche for persistent viral infection [13,14]. However, it is unknown whether this tropism for a relatively rare cell type regulates viral population diversity.

Type 2 cytokines such as IL-4 are well-established to induce tuft cell hyperplasia and enhance CR6 infection [13,44–47]. To assess the link between tuft cell infection and viral diversity, we treated WT and *Ifnlr1*[-/-] mice with two doses of recombinant IL-4 as described previously [13]. Mice were then infected with CR6[BC], and stool and tissues were analyzed at 5 dpi. We confirmed expected increases in expression of tuft cell markers *Dclk1* and *Cd300lf*, also the receptor for MNoV, with IL-4 treatment in both WT and *Ifnlr1*[-/-] mice in colon and ileum samples (**Fig 4A and 4B**). Consistent with prior studies, IL-4-treated mice exhibited significantly increased viral loads in stool from both WT and *Ifnlr1*[-/-] mice and WT colons, though viral levels in the ileum were not significantly enhanced (**Fig 4C**). With these increased viral levels, we observed enhanced barcode richness in the stool and colons of IL-4 treated mice of both genotypes, as well as in the ilea of *Ifnlr1*[-/-] mice (**Fig 4D**). These findings indicate that tuft cell abundance is an important bottleneck for CR6 population diversity in both tissues and fecally shed virus.

## Type I IFN, in combination with other IFNs, limits dissemination and extraintestinal viral diversity

Type I IFNs, which signal through receptor IFNAR1, are critical to limiting extraintestinal dissemination of CR6 [19,22]. Type II IFN, which acts through receptor IFNGR1, has been shown to play a supportive role to type I IFN signaling in regulating extraintestinal viral replication of other MNoV strains [48,49]. Transcription factor STAT1 relays signals for type I, II, and III IFNs, and is critical to limiting both intestinal (via type III IFN signaling), and extraintestinal (via type I and II IFN signaling) replication of CR6 [20–22,50,51]. To assess the role of IFN signaling in controlling viral population dynamics during extraintestinal viral dissemination, we orally infected WT, *Ifnar1*[-/-], *Ifnar1*[-/-]*Ifngr1*[-/-], and *Stat1*[-/-] mice with CR6[BC] and analyzed viral levels and barcode diversity at 5 dpi in stool, colon and spleen. As expected, we observed increased intestinal viral loads, reflected in stool and colon levels, of *Stat1*[-/-] mice with no significant changes in *Ifnar1*[-/-] or *Ifnar1*[-/-]*Ifngr1*[-/-] mice (**Fig 5A**). In contrast, both *Ifnar1*[-/-]*Ifngr1*[-/-] and *Stat1*[-/-] mice exhibited significantly increased viral levels in the spleen compared to WT and *Ifnar1*[-/-] (**Fig 5A**). While viral levels were not significantly enhanced in *Ifnar1*[-/-] mice, the likelihood of virus being detectable in the spleen was significantly increased (Fisher's exact test, p = 0.0081) (**Fig 5A**). Splenic viral levels in *Stat1*[-/-] mice were significantly higher than in *Ifnar1*[-/-]*Ifngr1*[-/-] mice (**Fig 5A**), cumulatively indicating that viral dissemination is limited by combinatorial activities of type I, II, and III IFN signaling, in holding with prior reports [21,22,49].

In stool and colon, consistent with the alterations in viral levels, *Stat1*[-/-] mice exhibited higher barcode richness while *Ifnar1*[-/-] and *Ifnar1*[-/-]*Ifngr1*[-/-] mice showed similar richness to WT mice, with the exception of a subtle increase in richness in *Ifnar1*[-/-] stool (**Fig 5B**). Results in *Stat1*[-/-] mice phenocopied our observations in *Ifnlr1*[-/-] mice (**Fig 2**), supporting that disrupted type III IFN signaling likely underlies the enhanced intestinal viral diversity in *Stat1*[-/-] mice. In the spleen, enhanced barcode richness was observed in *Ifnar1*[-/-], *Ifnar1*[-/-]*Ifngr1*[-/-], and *Stat1*[-/-] mice, consistent with the relative increases in viral dissemination to this extraintestinal site (**Fig 5B**). While loss of type II IFN signaling did not further enhance the barcode richness observed in *Ifnar1*[-/-] mice, splenic barcode richness was significantly higher in *Stat1*[-/-] mice (**Fig 5B**), indicating combinatorial roles for type I and III IFNs, with possible contribution from type II IFN, in limiting extraintestinal viral replication and thereby diversity.

## Viral dissemination can occur independently of intestinal replication in mice lacking type I IFN signaling

Next, to determine whether the systemic spread of CR6 required initial intestinal infection for subsequent dissemination or could occur independent of the intestine, we analyzed the

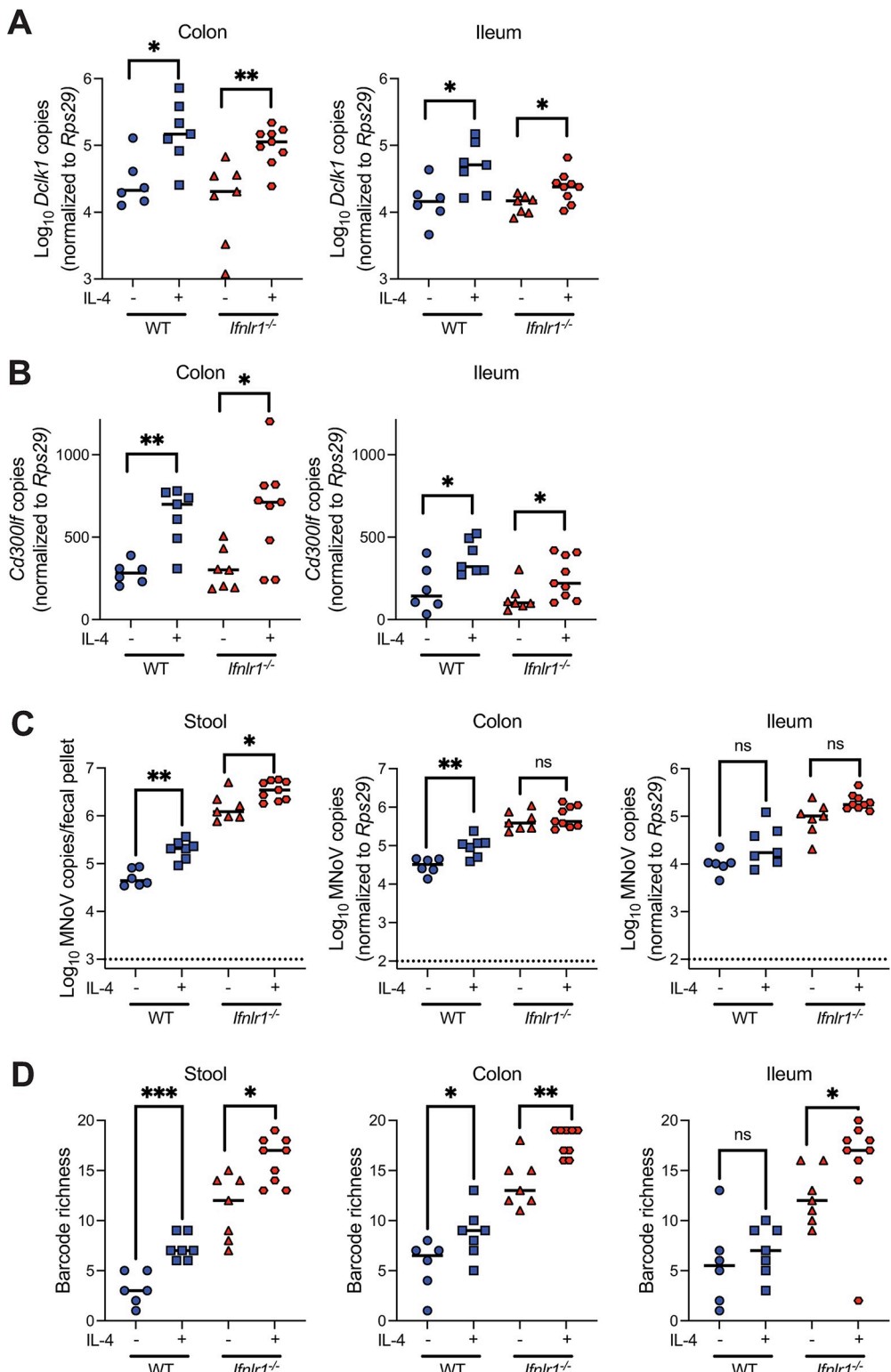

**Fig 4. Tuft cell abundance is a bottleneck for intestinal CR6 replication and viral diversity.** Across two independent experiments, WT (N = 7) and *Ifnlr1*⁻/⁻ (N = 9) mice were given two doses of r-IL-4 complexes at 48 and 24 h prior to infection, with untreated WT (N = 6) and *Ifnlr1*⁻/⁻ (N = 7) serving as controls. Stool and tissues were collected at 5 dpi. (**A,B**) Expression of tuft cell markers *Dclk1* (**A**) and *Cd300lf* (**B**) was quantified by qPCR at 5 dpi in colon and ileum samples. (**C**) Stool and tissue CR6 levels were quantified. (**D**) Barcode richness was evaluated from these same samples. Analyzed by Mann-Whitney test: *p < 0.05; **p < 0.01; ***p < 0.001; ns, not significant.

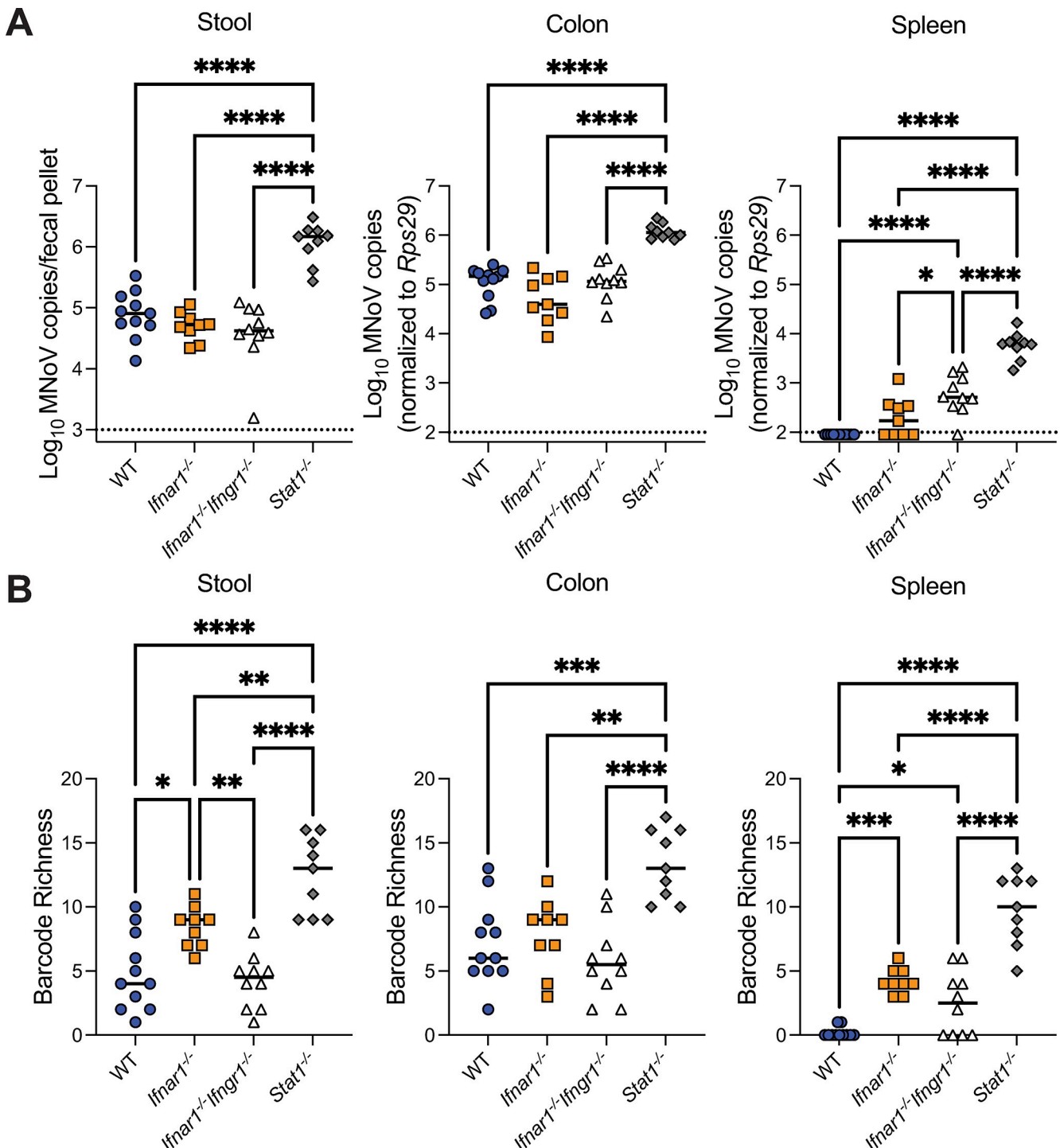

**Fig 5. Type I IFN, in combination with other IFNs, limits dissemination and extraintestinal viral diversity.** WT (N = 11), *Ifnar1*[-/-] (N = 9), *Ifnar1*[-/-]*Ifngr1*[-/-] (N = 10), and *Stat1*[-/-] (N = 9) mice were infected with CR6[BC] and samples were collected at 5 dpi. (**A**) Viral levels in stool, colon, and spleen were assessed by qPCR. (**B**) Barcode richness was evaluated from these same samples. Analyzed by one-way ANOVA with Tukey's multiple comparisons test: *p < 0.05; **p < 0.01; ***p < 0.001; ****p < 0.0001; ns, not significant.

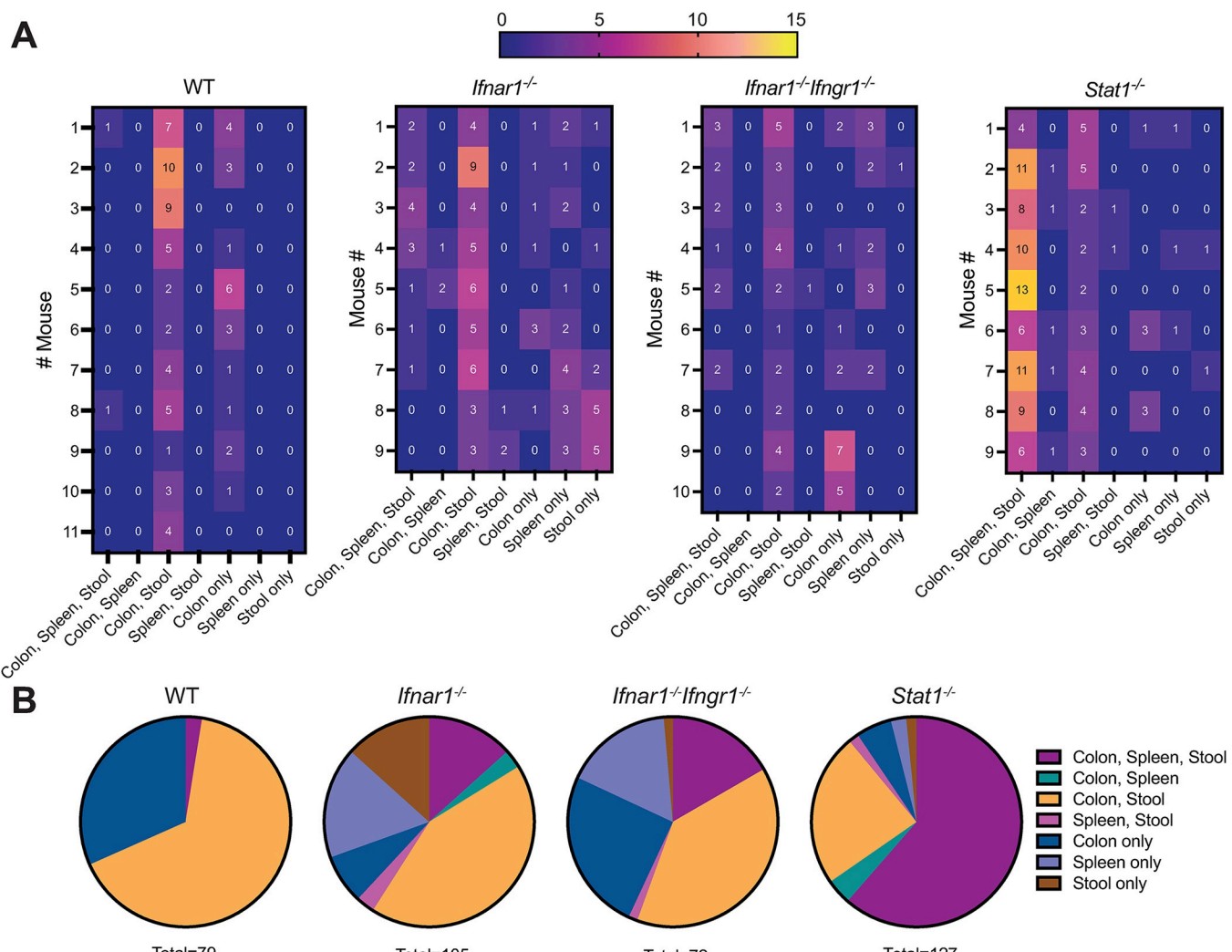

**Fig 6. Viral dissemination can occur independently of intestinal replication in mice lacking type I IFN signaling.** Barcode distribution in CR6^BC-infected WT, *Ifnar1^-/-*, *Ifnar1^-/-Ifngr1^-/-*, and *Stat1^-/-* mice at 5 dpi was analyzed for being unique or shared at different sites. (**A**) Heatmaps indicate the number of barcodes detected at the indicated unique or shared sites per mouse. (**B**) Pie-charts indicate the proportions of the total number of barcodes detected at the indicated unique or shared site, summed from all mice in the indicated groups.

distribution of barcodes across tissues in WT, *Ifnar1^-/-*, *Ifnar1^-/-Ifngr1^-/-*, and *Stat1^-/-* mice. In WT, *Ifnar1^-/-* and *Ifnar1^-/-Ifngr1^-/-* mice, barcodes were predominantly shared between colon and stool (**Fig 6A and 6B**), consistent with our earlier findings (**Fig 3**). Intriguingly, analysis of splenic barcode distribution in *Ifnar1^-/-* and *Ifnar1^-/-Ifngr1^-/-* mice revealed that a unique barcode was equally likely to be detected in the spleen alone (~17% for both genotypes) as it was to be found in the colon, spleen and stool (~13% for *Ifnar1^-/-* and ~17% for *Ifnar1^-/-Ifngr1^-/-* mice) (**Fig 6A and 6B**). These data indicate that, in the absence of type I IFN signaling, viral dissemination can likely occur both from intestinally replicated virus as well as independently of intestinal replication. In *Stat1^-/-* mice, wherein barcode richness was much higher at all sites tested, most barcodes were shared between all sites with some barcodes exclusively present in colon and stool (**Fig 6A and 6B**). Thus, these data support a model in which type I IFN signaling limits dissemination of virus that is not replicating in the intestine, but in the absence of

signaling by all IFNs, unrestricted viral replication leads to dramatic increases in viral diversity that are shared broadly.

## Discussion

Host immune and tropism barriers influence viral pathogenesis by modulating viral population dynamics, serving as bottlenecks during viral replication and spread within the host. To date, the role of these barriers in regulating the viral diversity of NoV infection has not been explored. In this study, the application of genetic barcoding of CR6 permitted us to carefully define both how these bottlenecks limit viral diversity at different sites and revealed new insights about the source(s) of shed and disseminated virus. The insertion of the small barcode showed no effect on viral fitness *in vitro* or *in vivo*, and barcodes were maintained in the virus over the course of *in vitro* passaging and *in vivo* longitudinal infection.

In WT mice, CR6 establishes asymptomatic infection exclusively in short-lived tuft cells, from which high levels of infectious virus are persistently shed [13,31,44]. Only a small number of unique barcodes were observed per mouse even at early timepoints, indicating that initial infection events are limited in WT mice. We also observed longitudinal loss of intestinal and shed viral population diversity over the course of a 21-day infection, potentially reflecting either turnover of infected tuft cells or effects of adaptive immune responses in limiting viral diversity by targeting a subset of infected cells. Infection in WT mice is limited to the intestine, and analysis of barcodes in ileal and colonic tissues, the major sites of viral infection, revealed distinct barcodes present at these sites, suggesting independent infection events of ileal and colonic tuft cells. Further, analysis of the proportions of shared barcodes between these tissues with stool revealed that the colon is the dominant contributor to shed virus compared to the ileum. This observation may reflect distinct characteristics of infected tuft cells between these intestinal tissues (e.g. differential capacity to be infected or to produce or package virus) or a relative benefit to colon-derived virus in transit through the intestine for fecal shedding.

Because IFN-λ is a well-established intestinal regulator of CR6, particularly at acute timepoints [18,21,22,24], we evaluated its role in regulating viral diversity. We found that the increased intestinal viral replication seen in *Ifnlr1*$^{-/-}$ mice was associated with a substantial increase in barcode richness, supporting that IFN-λ serves to limit initial tuft cell infection events and therefore overall viral diversity. We also determined that STAT1 is a critical transducer of IFN-λ signaling to regulate intestinal viral diversity. In *Ifnlr1*$^{-/-}$ mice, longitudinal viral diversity was also lost, and indeed by 21 dpi barcode richness was equivalent between WT and *Ifnlr1*$^{-/-}$ mice, indicating that IFN-λ has an active role in limiting diversity only at acute timepoints.

Having found that IFN-λ is such a critical bottleneck at early timepoints for viral diversity, we were also curious whether CR6's tropism for rare tuft cells (~0.5% of intestinal epithelial cells [52]) served as an additional limitation for initial infection events. We administered IL-4 to mice prior to infection, a treatment well-established to enhance intestinal tuft cell numbers [13,44–47], and found that this enhanced infection and barcode richness in WT and *Ifnlr1*$^{-/-}$ mice. Remarkably, almost all barcodes were recovered in *Ifnlr1*$^{-/-}$ mice treated with IL-4, supporting combinatorial and critical roles for limited tuft cell numbers and innate immune signaling in restricting intestinal viral diversity.

Finally, we leveraged our barcoded viral library to explore viral dissemination to systemic tissues in *Ifnar1*$^{-/-}$, *Ifnar1*$^{-/-}$*Ifngr1*$^{-/-}$, and *Stat1*$^{-/-}$ mice. We confirmed prior findings that type I and II IFNs are dispensable for regulating viral levels in the intestine at acute timepoints [21], with concomitant findings for viral diversity. Because type I IFN signaling is an established regulator of extraintestinal MNoV dissemination [19,22], we evaluated viral levels in the

spleen, which showed that type I and III IFNs play combinatorial roles in restricting viral levels as well as viral diversity at this site. In *Ifnar1*[-/-] and *Ifnar1*[-/-]*Ifngr1*[-/-] mice, wherein intermediate levels of barcodes were observed in the spleen, we were able to determine that unique barcodes were present in the intestine and the spleen, supporting that MNoV is disseminated via intestinal replication-dependent and independent routes. IFN signaling is known to restrict MNoV immune cell infection [50, 53]. Thus, it is possible that in IFN signaling-deficient mice, a combination of immune cells and tuft cells are initially infected, with early systemic trafficking of infected immune cells followed by dissemination of tuft cell-derived virus, permitting distinct barcode populations with either exclusive splenic detection or overlapping colonic and splenic detection. In *Stat1*[-/-] mice, wherein enhanced intestinal and systemic viral replication are both present, most barcodes are present at all sites likely due to increased dissemination of intestine-derived virus.

While here, a focused pool of 20 barcoded viruses was sufficient to permit numerous inferences regarding viral source(s) for shed and systemic virus, future studies would likely benefit from a richer pool of barcodes, as well as analyses of additional early timepoints, to permit finer resolution of acute infection events. We also cannot exclude the possibility of barcodes below the limit of detection in our assay contributing to viral replication. Despite these limitations, our study revealed that innate immune signaling and tuft cell availability account for nearly all the viral diversity bottlenecks observed in WT mice, consistent with their roles in regulating viral replication and dissemination. Further, we found that the colon is the main source of fecal virus despite detectable small intestinal infection, and that virus may bypass the intestine for systemic spread in the absence of type I IFN signaling. This study thus provides important new insights into the mechanisms of NoV pathogenesis.

## Supporting information

**S1 Fig. Barcoded CR6 (CR6**[BC]**) shows similar infectivity to CR6 and maintains barcodes *in vitro* and *in vivo*.** (**A**) The genome copy:plaque forming unit (PFU) ratio for the 20 individual barcoded viruses was assessed in comparison to the relative proportion of each barcoded virus in the inoculum. (**B,C**) *Cd300lf*[-/-] mice (N = 5) were inoculated with CR6[BC] and stool viral shedding at 3 and 5dpi (**B**) and tissue viral levels at 5dpi (**C**) were assessed using qPCR. (**D,E**) Barcode richness of *Cd300lf*[-/-] stool (**D**) and tissues (**E**) was determined. (**F**) The proportion of mice (N = 16 total including WT and *Ifnlr1*[-/-] mice depicted in Fig 2) with each individual barcode detected at any time between 5-21dpi was compared to the level of that barcode in the inoculum. (**G**) The proportion of reads in the indicated tissues maintaining barcodes was assessed at 21dpi. Results were analyzed by Mann-Whitney test. ns, not significant.
(DOCX)

**S1 Table. List of the sequences containing 6 nucleotide barcodes (underlined) used in the study.** Sequences in red were random sequences used for the initial validation of the sequencing run and analysis; the last sequence (26) is the CR6 viral sequence with no barcode.
(DOCX)

**S2 Table. Sequencing reads obtained for individual barcodes in inoculum and ileum, colon or stool of WT or *Ifnlr1*[-/-] mice at 21dpi, or longitudinally in stool at 5, 14, and 21dpi (with limit of detection of ≥ 10).** Colored boxes indicate reads between 0–9. Font color indicates inoculum used in mice, either #1 or #2.
(XLSX)

**S3 Table. Sequencing reads obtained for individual barcodes in inoculum and ileum, colon, or stool samples for IL-4-treated or untreated WT and *Ifnlr1*[-/-] mice at 5 dpi with**

**limit of detection of ≥ 10.** Colored boxes indicate reads between 0–9 reads.
(XLSX)

**S4 Table. Sequencing reads obtained for individual barcodes in inoculum or spleen, colon or stool samples of WT,** *Ifnar1*-/-, *Ifnar1*-/-*Ifngr1*-/-, *or Stat1*-/- **mice at 5dpi with limit of detection of ≥ 10.** Colored boxes indicate reads between 0–9 reads. Font color indicates inoculum used in mice, either #1 or #2.
(XLSX)

**S1 Data. Data that underlies Fig 1 of this paper.**
(XLSX)

**S2 Data. Data that underlies Fig 2 of this paper.**
(XLSX)

**S3 Data. Data that underlies Fig 3 of this paper.**
(XLSX)

**S4 Data. Data that underlies Fig 4 of this paper.**
(XLSX)

**S5 Data. Data that underlies Fig 5 of this paper.**
(XLSX)

**S6 Data. Data that underlies Fig 6 of this paper.**
(XLSX)

## Author Contributions

**Conceptualization:** Forrest C. Walker, Broc T. McCune, Megan T. Baldridge.

**Data curation:** Somya Aggarwal, Xiaofen Wu, Megan T. Baldridge.

**Formal analysis:** Somya Aggarwal, Forrest C. Walker, James S. Weagley, Xiaofen Wu, Dylan Lawrence, Megan T. Baldridge.

**Funding acquisition:** Megan T. Baldridge.

**Investigation:** Somya Aggarwal, Forrest C. Walker, Broc T. McCune, Lawrence A. Schriefer, Heyde Makimaa, Pratyush Sridhar.

**Methodology:** Somya Aggarwal, Forrest C. Walker, Broc T. McCune, Lawrence A. Schriefer, Pratyush Sridhar.

**Project administration:** Megan T. Baldridge.

**Software:** Forrest C. Walker, James S. Weagley.

**Supervision:** Megan T. Baldridge.

**Visualization:** Somya Aggarwal, Megan T. Baldridge.

**Writing – original draft:** Somya Aggarwal, Megan T. Baldridge.

**Writing – review & editing:** Somya Aggarwal, Forrest C. Walker, James S. Weagley, Broc T. McCune, Xiaofen Wu, Lawrence A. Schriefer, Heyde Makimaa, Dylan Lawrence, Pratyush Sridhar, Megan T. Baldridge.

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
