## [Decision Letter · Decision Letter 0]

27 Feb 2024

Dear Dr. Baldridge,

Thank you very much for submitting your manuscript "Interferons and tuft cell numbers are bottlenecks for persistent murine norovirus infection" for consideration at PLOS Pathogens. As with all papers reviewed by the journal, your manuscript was reviewed by members of the editorial board and by several independent reviewers. The reviewers appreciated the attention to an important topic. Based on the reviews, we are likely to accept this manuscript for publication, providing that you modify the manuscript according to the review recommendations.

Sincerely,

George A. Belov, PhD

Academic Editor

PLOS Pathogens

Guangxiang Luo

Section Editor

PLOS Pathogens

Michael Malim

Editor-in-Chief

PLOS Pathogens

orcid.org/0000-0002-7699-2064

Reviewer Comments (if any, and for reference):

Reviewer's Responses to Questions

**Part I - Summary**

Reviewer #1: This report by Aggarwal and colleagues, seeks to understand the consequences of host innate immunity on norovirus population dynamics. Using the murine norovirus (MNV) model system, the authors generate a pool of 20 different MNV clones each containing a genetic barcode by inserting 6-nucleotoide sequences at the 3’ end of the viral NS4 coding sequence. Using this pool of barcoded MNV the authors find that persistent infection involves a series of reinfection events and genetic richness decreased over time, with IFN lambda receptor knockout mice, viral richness and replication in the intestines was increased. Increasing host tuft cell numbers by IL-4 treatment increased barcode richness suggests tuft cell numbers are a bottle neck during MNV infection. Finally, Stat1 knockout mice showed increased viral diversity at extraintestinal sites indicating different interferons as critical bottlenecks at tissue types, and that disseminated splenic viruses were distinctly barcoded from intestinal viruses. This greater understanding of how host cell immunity influences norovirus population dynamics is original and of high importance to researchers in this field and likely of a broader interest to those studying related viruses (in particular human norovirus).

Generally the manuscript is very well written and described, the methodology is overall rigorous and experiments well controlled. The authors conclusions I believe are supported by the evidence provided. There are some additional data that would be informative to the study, in addition to some minor changes to help readability.

Reviewer #2: In this well written and straightforward manuscript, the authors describe their creation of a panel of 20 barcoded viruses using the persistent murine norovirus strain, CR6. After measuring in vitro replication efficiency, the authors infected this virus pool into wild type and transgenic mice and quantified these viruses in intestinal tissue, the spleen and the stool. The barcodes allowed the authors to also evaluate virus populations in these tissues. The major conclusions drawn by the authors from their results include: 1) virus is predominantly shed in stool comes from viruses that are infecting the colon, 2) that viral barcode richness decreases over time in all tissues and in all mice, regardless of immune status, 3) but that richness is increased in IFN-lambda mice compared to all others and that this increase in richness is due to improved viral replication; 4) increases in the number of tuft cells (induced by IL-4 treatment) increases richness, likely also due to improved replication resulting from the increased availability of tuft cells; 5) loss of IFN-I signaling increased viral diversity in extraintestinal sites indicating that IFN-I is responsible for limiting spread of MNV; 6) that intestinal viruses were different from extraintestinal viruses. The data presented in the manuscript provides important insight into norovirus replication, shedding and dissemination and the tool of barcoded viruses created to do these studies will likely prove to be extremely valuable in discerning other aspects of norovirus pathogenesis and replication. While the results presented in the manuscript are significant, there are some concerns which should be addressed to strengthen the results and improve the clarity of the overall conclusions.

Reviewer #3: The manuscript titled “Interferons and tuft cell numbers are bottlenecks for persistent murine norovirus infection” submitted for publication by Aggarwal et al describes the use of a barcode-based technology to study norovirus pathogenesis in a murine model. This paper addresses unanswered questions surrounding host factors, such as epithelial cell types and immune signaling pathways that drive norovirus population dynamics. The manuscript is extremely well written, and the experiments are well designed, executed, and validated with controls and statistics including upfront validation of the bar-coded viral strains compared to wild type strains. In general, the conclusions are appropriate based on the data presented. The bar code technology has been used with other viral systems, and the authors capitalize on its availability to adapt it to the murine norovirus model. New discoveries include the concept that IFNg is a signaling bottleneck for intestinal murine norovirus replication, the colon as the major contributor to stool shedding of murine norovirus, the presence of the tuft cell as a bottleneck for intestinal murine norovirus replication, and the role of Type I IFN and not intestinal viral replication in limiting extraintestinal spread of murine norovirus. These findings are interesting and provide substantial insight into knowledge of murine norovirus replication, extra-intestinal spread, and pathogenesis that warrants publication in PLOS Pathogen. The discussion of the paper seems a bit lacking on integrating these findings into the bigger picture of the pathogenesis of other murine norovirus strains (ie are these findings only specific to the CR6 strain) and more importantly, if any of the findings relate to human norovirus pathogenesis.

**Part II – Major Issues: Key Experiments Required for Acceptance**

Reviewer #1: (No Response)

Reviewer #2: (No Response)

Reviewer #3: • The percentage of barcode recovery in Fig 1D seems to vary from close to 0% all the way to about 12%. If each barcode is used in equal proportions, how is this difference explained?

• Re the conclusion in lines 266-268, What is the limit of detection of the bar codes. Is it possible that bar coded viruses are contributing to the overall viral load but are below the limit of detection of the approach?

• There seems to be variability between figures 2 and figures 4 and 5 in terms of the barcode richness in the controls. Is this difference statistically significant and what is the relevance of this variability to the findings and conclusions?

**Part III – Minor Issues: Editorial and Data Presentation Modifications**

Reviewer #1: 1. The “barcodes” used are inserted after nucleotide 2601 which lies at the C-terminal end of NS4. It would be informative to provide additional information on the barcodes used and how they alter amino acid sequence, where they lie in respect to the NS4NS5 junction and if they alter processing of the polyprotein. Additional methodological information will allow other researcher studying MNV and other viruses to use a similar approach.

2. The authors show CR6BC is sequence stable in vitro for 72 hours however several of the mice experimental groups are upto 21 days post infection. Are the barcodes maintained without change or loss in vitro or in vivo upto 3 weeks. Likewise, is there any difference or variations in the replication competency of individual barcoded viruses? Differences or reduced replication competency many be expected since the original transposon mutagenesis paper where this insertion was characterised (Thorn et al 2012) suggested insertions after nucleotide 2600 cause ~2-log reduction in viral replication.

3. In Figure 1D, it was unclear why the relative proportions of sequence reads for each barcoded virus are different when the same number of PFU were pooled to generated CR6BC. Does this represent bias in the sequence data or different PFU:particle ratios?

4. Lines 219:220: Here the manuscript says “distinct barcodes were observed in each mouse, indicated that this low number was not secondary to poor viability of some clones”. It is hard for the reader to conclude this from the data presented in Fig 2B, it could be that one or two clones are poorly viable even if different barcodes were prevalent in each mouse/sample. Were there any statistically significant differences in the barcodes represented which may suggested poorer viability.

5. The choice of colours and symbols for some figures (e.g., Figure 2) makes it harder to interpret for some readers with types of colour blindness. Could more distinct colours and/or symbols be used to make all the figures for accessible.

6. Figure 5: Here the authors have conducted a thorough statistical comparison, however, is a comparison of all groups to each other required. Could the figure be simplified by only presenting key statistical comparisons or only highlighting the significant comparisons.

Other minor comments

7. Line 94: Define VP-SFM

8. Line 141: For consistency uL not ul

9. Line 153: Subscript 4 i.e., MgSO4

10. Line 184: NS4 not p18

Reviewer #2: Lines 189-190. The barcoded viruses are not in equivalent numbers based on in vitro work. While no single virus makes up more than 15% of the population it appears some viruses make up less than 1% of the pool. When in vitro work was repeated, was this stratification of population consistent? Did some viruses always become predominant? Was it always the same viruses? Do the authors have an explanation for why, if equal PFUs were added to the pool, some barcodes drop below 1% of the population? Does this indicate that some of the viruses in the pool have growth defects which could impact interpretation of in vivo results? It is stated on line 219 that the in vivo data indicates there is no differences in clone viability, but that specific barcode data is not provided (Fig 2).

Lines 219-220. Since the authors possess data showing the distinct barcodes at each site, including that data in the supplemental section would be helpful and also possibly provide insight into any issues with in vivo replication of specific viral barcodes.

Lines 231-233. Which barcodes are lost during persistent infection? Is it consistent between mice? Including a table with this data in the supplemental section would be helpful.

Lines 276-277. This conclusion takes into consideration only the likelihood test mentioned immediately prior while ignoring the rest of the data presented in this section showing that the method used cannot sufficiently determine if viruses shed in stool comes from ileal virus or colonic virus. The statement should be revised to accurately reflect all the data presented in this section.

Lines 291-304. The authors state that tufts cells are the reservoir for persistent virus infection, yet the data displayed in this section only examines 5 dpi (previously indicated by the authors as the “acute timepoint”. How did the increase in tufts cells alter diversity at the 21 dpi, persistent timepoint? Also, it is not stated how the authors validated that IL-4 treatment increased the abundance of tufts cells and how large was that increase. If not possible to perform, the authors should include typical tuft cell number increases.

Lines 344-346. As indicated previously by the authors, barcode richness appears to be tied to viral replication. Therefore, do the Type I and Type III IFNs actually and directly control viral richness as the authors conclude? Or do they rather have an indirect effect related to their ability to regulate viral replication and dissemination? Acknowledgement and expansion of this point in the manuscript would be helpful.

Lines 359-373 . Why did the authors not include ileal barcodes in their comparison with the spleen? Given that the ileum contains the majority of Pyers Patches and the connection of PPs to the MLNs which drain to other lymphatic tissues, could there be a correlation between ileal replication and migration to the spleen? This data is very related to the conclusions the authors are trying to draw. Even including data from WT mice would be helpful (in the case that ileal tissue wasn’t collected for the transgenic mice).

Reviewer #3: • In figure 1, it would improve the flow of the paper to have the proportion of barcoded viruses in the stool and tissue to go along with Fig 1E/F. Splitting into two figures with one on the BV2 cells and one on the mouse model would be good.

• In lines 220-222, the conclusion seems overstated. It could just be random chance that specific bar codes were observed rather that the fact that a limited number of unique viruses could establish/maintain infection.

• In Fig. 2, it is hard to tell the triangles apart. The use of a different shape/color would improve the readability of this figure.

• In Fig 2C, the data is shown as MNoV copies per fecal pellet but there is no description in the materials and methods as to how this was collected. A random sample from the cage versus one taken directly from the mouse at the same time might have totally different levels of virus.

• In lines 263-264, how do the authors imagine that colonic infection contributes to ileal infection? Does CR6 cause viremia?

• Re the conclusion in lines 276-277, how many tuft cells are present in the mouse colon compared to the small intestine that would support this conclusion?

• In lines 407-410, it should

---

## [Editor Report · Decision Letter 1]

20 Apr 2024

Dear Dr. Baldridge,

We are pleased to inform you that your manuscript 'Interferons and tuft cell numbers are bottlenecks for persistent murine norovirus infection' has been provisionally accepted for publication in PLOS Pathogens.

Best regards,

George A. Belov, PhD

Academic Editor

PLOS Pathogens

Guangxiang Luo

Section Editor

PLOS Pathogens

Michael Malim

Editor-in-Chief

PLOS Pathogens

orcid.org/0000-0002-7699-2064
---

## [Editor Report · Acceptance letter]

29 Apr 2024

Dear Dr Baldridge,

We are delighted to inform you that your manuscript, "Interferons and tuft cell numbers are bottlenecks for persistent murine norovirus infection," has been formally accepted for publication in PLOS Pathogens.

Best regards,

Michael Malim

Editor-in-Chief

PLOS Pathogens

orcid.org/0000-0002-7699-2064